# Recovery of Functional Proteins from Pig Brain Using pH-Shift Processes

**DOI:** 10.3390/foods11050695

**Published:** 2022-02-26

**Authors:** Jaruwan Chanted, Worawan Panpipat, Ling-Zhi Cheong, Manat Chaijan

**Affiliations:** 1Food Technology and Innovation Research Center of Excellence, School of Agricultural Technology and Food Industry, Walailak University, Nakhon Si Thammarat 80160, Thailand; jaruwanchanted@gmail.com (J.C.); pworawan@wu.ac.th (W.P.); 2Zhejiang-Malaysia Joint Research Laboratory for Agricultural Product Processing and Nutrition, College of Food and Pharmaceutical Science, Ningbo University, Ningbo 315211, China; cheonglingzhi@nbu.edu.cn

**Keywords:** pH-shift, protein isolate, pig brain, by-product, functionality

## Abstract

The goal of this work is to explore if pH-shift processing could be used as a cold refinery technique to manufacture pig brain protein isolate (PI). Pig brain protein had the highest solubility at pH 2 (acid method) and pH 12 (alkaline method). As the protein solution’s zeta-potential was near 0 with the lowest solubility, pH 5.0 was chosen as the precipitation pH. Alkaline process produced a 32% dry matter yield with phospholipid content of 35 mg/100 g. The alkaline-made PI was better at forming soft gels and had good emulsifying and foaming capabilities. Although the acid-made PI included less residual lipid and total haem protein and was whiter in colour, it could not be gelled. Acid-made PI was more prone to lipid oxidation with a poorer ability to function as an emulsifier and foaming agent. Thus, functional proteins from pig brain may be isolated using the alkaline pH-shift technique.

## 1. Introduction

The issue of food sustainability and food security has existed among human societies due to an expanding worldwide population, and an increase in food demand is projected by 2050, leading to a lack of animal-based protein supply from farmed livestock [1]. The world’s food demand will be over 60% higher by 2050 than it is today [2]. As a consequence, researchers have been looking for a sustainable food supply chain [3]. Proteins are indeed an essential ingredient for life and provide technological utility to food products. One of the solutions for this is to fully utilise protein-containing by-products from livestock slaughtering and processing.

In 2020, Thailand’s pig production is expected to exceed 20.5 million heads, with pork consumption expected to total 1.3 million tons [4]. Increased pork consumption produces by-products such as blood, bone, bristle, fat trimming, viscera, and brain [5,6,7]. Between 60 and 70% of the slaughtered carcass is made up of by-products, with around 40% edible and 20% inedible [7]. Some of these by-products are traditionally used in some countries around the world in a variety of recipes [6] and can be effectively value-added using additional processes such as thermal, chemical, centrifugation, washing, and combined processes to produce lard, flavour concentrate, plasma, red blood cells, gelatin, protein hydrolysates, and others [7,8,9]. However, a range of factors, such as religion, culture, income, and personal taste, have an impact on the utilisation of meat by-products. Various meat by-products can be deemed edible in some areas but inedible in others, depending on the region and local traditions. In actuality, some countries use high-nutrient by-products such as heart, liver, blood, lung, spleen, kidney, tripe, and brains in their cuisines [6]. Despite the fact that pig brain is a common by-product of slaughtering and pork processing, it has yet to be widely employed, particularly for human consumption [10]. Furthermore, there is no academic understanding on how to increase the value of pig brain. Soup, gravy, stew, curry, and fried meals have been identified as the principal methods of employing pig brain.

The effectiveness of the pH-shift approach to isolate protein from a variety of sources has been well-reported [11,12,13,14,15]. Isoelectric (pI) precipitation commonly follows protein solubilisation at the proper acid or alkaline pH to recover a functional protein isolate in the pH-shift process [15]. Despite previous research employing the pH-shift method to separate protein from fish, numerous attempts to use the pH-shift approach to recover protein from diverse biomaterials have been undertaken [1,12,16]. The pH-shift process, according to the findings, might be used to isolate protein from any source of protein. The pH-shift methodology has a number of benefits over typical recovery methods for extracting protein isolates from a variety of sources, including better yields, improved techno-functional properties, and greater impurity removal efficiencies [14]. Furthermore, no pH-shift-made protein isolate (PPI) from pig brain has been reported. Using acid and alkaline solubilisation procedures, this work aims to generate and characterise PPI from pig brain as a potential food ingredient. The quality features and functional aspects of acid-made protein isolate (Acid-PPI) and alkaline-made protein isolate (Alk-PPI) were compared—particularly, rheological, gelling, emulsifying, and foaming properties. The discoveries could help boost pig brain exploitation in the sustainable meat industry, allowing for the zero-waste concept to be realised.

## 2. Materials and Methods

### 2.1. Chemicals

All chemicals and reagents used in this study, such as ammonium thiocyanate, bromophenol blue sodium salt (BPB), chloroform, ferric chloride hexahydrate, hydrochloric acid (HCl), methanol, sodium dodecyl sulfate (SDS), and sodium hydroxide (NaOH), were analytical grade and purchased from Sigma-Aldrich (St. Louis, MO, USA).

### 2.2. Collection and Preparation of Pig Brains

Thirty brains of crossbred pigs (Landrace × Large white × Duroc, LLD) at 4 months of age were collected from the Shaw Processing Food. Co., Ltd. in Nakhon Si Thammarat, Thailand. The brains came from healthy pigs approved by Thailand’s Bureau of Livestock Standards and Certification. Within 1 h, the obtained samples were delivered to Walailak University’s Laboratory in ice with a sample-to-ice ratio of 1:2. The brains were separated into 3 groups (each with 10 brains; *n* = 3). The brains were then rinsed in cold water (4 °C), drained, and chopped using a Talsa Bowl Cutter K15e (The Food Machinery Co., Ltd., Kent, UK) to create a homogeneous composite sample.

### 2.3. Profiles of Zeta-Potential, Solubility, Colour, and Total Haem Protein Content of Pig Brain Proteins as Affected by pH Adjustment

An IKA homogeniser (Model T25 digital Ultra-Turrax, Staufen, Germany) was used to homogenise ground pig brain (100 g) with 900 mL cold distilled water (4 °C) for 5 min at 20,000 rpm [1,15]. A Cyberscan 500 pH meter (Eutech, Singapore) was used to examine the pH of the homogenate, which was adjusted to 1–14 using either 1 M HCl or 1 M NaOH. The mixture was centrifuged at 8500× *g* for 20 min at 4 °C and the supernatant was subjected to assays of zeta-potential, protein content, colour, and total haem protein content.

The Zetasizer Nano-ZS90 (Malvern Instruments Ltd., Worcestershire, UK) was used to investigate the zeta-potential. The Biuret method [17] was used to determine the protein content, and the protein solubility was calculated in g per 100 g of sample. Colourimetric values, including *L** (lightness), *a** (redness/greenness), and *b** (yellowness/blueness) were analysed using a Hunterlab ColourFlex^®^EZ instrument (Hunter Assoc. Laboratory; Reston, VA, USA). The total haem protein was assessed by Chaijan and Undeland’s method [18] and stated in g of haemoglobin per 100 g of sample.

The pH with the highest protein solubility was identified as the solubilisation pH, whereas the pH with the lowest protein solubility was defined as the precipitation pH for the pH-shift procedure.

### 2.4. Acid and Alkaline pH-Shift Procedures

The PPI was produced using the method of Chaijan et al. [1]. The aqueous homogenate of ground pig brain prepared as mentioned above was adjusted to either pH 12 (alkaline method) or pH 2 (acid method) using 2 M NaOH or 2 M HCl with steady stirring (100 rpm/10 min). After centrifuging at 8500× *g* (20 min/4 °C), the supernatant was filtered through three layers of cotton gauze. The proteins were then precipitated by changing the pH to 5.0, then centrifuging under the same conditions as before. Without any more pH modifications, the precipitate, termed as PPI, was taken, weighed, and analysed.

### 2.5. Determination of Moisture Content and Dry Matter Yield 

The Acid-PPI and Alk-PPI were analysed for moisture content [19] and the dry matter yield of the PPI were calculated based on the initial dry matter content in the ground pig brain. 

### 2.6. Determination of Total Lipid, Phospholipid, and Haem Protein Contents

Bligh and Dyer’s method [20] was utilised to extract lipid from the PPI and recorded as g/100 g PPI. The phospholipid content of the extracted oil was analysed by a modified Stewart method [21] and expressed in mg/100 g PPI. 

The total haem protein content was assessed by the method of Chaijan and Undeland [18] and recorded as mg haemoglobin/g PPI. A sample and 3 volumes of 0.1 M phosphate buffer, pH 7, containing 5% SDS (*w*/*v*) was homogenised at 13,500 rpm for 20 s. The homogenate was heated in a water bath (85 °C) for 1 h and cooled under running tap water for 10 min. The solution was then centrifuged (5000× *g* /15 min/ 25 °C). The absorbance of the supernatant was read at 535 nm using a UV-Vis spectrophotometer with phosphate buffer as a blank. A standard curve of bovine haemoglobin (0–20 µM) was used.

### 2.7. Determination of Colour 

*L**, *a**, and *b** of the Acid-PPI and Alk-PPI were determined using a Hunterlab ColourFlex^®^EZ instrument. The following formula was used to determine the whiteness [15]:Whiteness = 100 − [(100 − *L**)^2^ + *a**^2^ + *b**^2^]^1/2^(1)

### 2.8. Determination of Hydrophobicity (HBP)

The HPB of PPI was determined using the approach of Somjid et al. [22]. Two hundred microlitres of 1 mg/mL aqueous BPB was added to 1 mL of PPI solution in 20 mM phosphate buffer (pH 6) and thoroughly mixed. PPI was not used in the control group. At room temperature (25–29 °C), samples and controls were agitated for 10 min. After centrifugation at 2000× *g* (15 min/room temperature), the absorbance of the supernatant was read at 595 nm (A) against a phosphate buffer blank using a Shimadzu UV-2100 spectrophotometer (Shimadzu Scientific Instruments Inc., Columbia, MD, USA). The content of BPB bound was calculated using the following formula:(2)$$\mathrm{BPB}\;\mathrm{bound}\;\left(\mu g\right)=200\;\mu g\times \left[\frac{A_{\mathrm{control}}-A_{\mathrm{sample}}}{A_{\mathrm{control}}}\right]$$

### 2.9. Determination of Rheological Properties

The rheological properties of PPI with the same moisture content of 91.12% were determined using a Rheometer (HAAKE MARS 60, Thermo Fisher Scientific Inc, Yokohama, Japan) [22]. When the temperature was pushed from 10 to 90 °C at a rate of 2 °C/min, changes in rheological parameters such as elastic modulus (G’), viscous modulus (G”), and tan δ were measured.

### 2.10. Determination of Gelling Properties

The method of Panpipat et al. [23] was used to make thermally induced PPI gels (91.12% moisture/2.5% NaCl). Setting took place at 40 °C/30 min, followed by cooking at 90 °C/20 min. The gels were chilled in ice water/30 min, then stored at 4 °C/24 h before being analysed. Breaking force, deformation, expressible drip, whiteness, texture profile analysis (TPA), and microstructure of the gel were all examined [23]. The determination of thiobarbituric-acid-reactive substances (TBARS) was carried out using the method of Buege and Aust [24]. The TBARS were measured in milligrams of malondialdehyde equivalents/kg sample.

### 2.11. Analyses of Interfacial Properties 

The emulsifying and foaming characteristics of Acid-PPI and Alk-PPI were determined in comparison with those of commercial soy protein isolate (SPI), whey protein isolate (WPI), and fresh egg albumen. Emulsion activity index (EAI), emulsion stability index (ESI), foam ability, and foam stability of the PPI were estimated using the method described by Panpipat and Chaijan [16].

For emulsifying properties, 2 mL of soybean oil was homogenised with 6 mL of protein solution (10 mg/mL) at 20,000 rpm for 1 min using an IKA homogenizer. At 0 and 15 min, emulsions were collected and diluted with 0.1% (*w*/*v*) SDS. The absorbance was then measured at 500 nm. EAI and ESI were calculated using the following formulas:(3)$$\mathrm{EAI}\;\left(m^{2}/g\right)=\frac{2\times 2.303\times A\times \mathrm{DF}}{l\emptyset C}$$
where A = absorbance at 500 nm, DF = dilution factor, l = path length (m), ø = oil volume fraction, and C = protein concentration (g/m^3^).
(4)$$\mathrm{ESI}\;\left(\min\right)=\frac{A0}{\Delta A}\times \Delta t$$
where A_0_ = absorbance at 500 nm (0 min), ΔA = A_0_- absorbance at 500 nm for 15 min, and Δt = 15 min.

Figure 1 (*w*/*v*) protein concentration was transferred into 100 mL cylinders. The mixtures were homogenised for 1 min at 13,400 rpm at room temperature. The sample was left out for 0, 30, and 60 min. The following equations were used to determine foam ability and foam stability:(5)$$\mathrm{Foam}\;\mathrm{ability}\;\left(\%\right)=\frac{\mathrm{VT}}{V0}\times 100$$
(6)$$\mathrm{Foam}\;\mathrm{stability}\;\left(\%\right)=\frac{\mathrm{Vt}}{\mathrm{VT}}\times 100$$
where VT is total volume after whipping, V0 is the original volume before whipping, and Vt is total volume after leaving at room temperature for 30 or 60 min. 

### 2.12. Statistical Analysis

All of the experiments were performed in triplicate (*n* = 3). The data were subjected to an ANOVA analysis. The means were compared using Duncan’s multiple range test. A *T*-test was employed to compare pairs of data. SPSS 23.0 (SPSS Inc., Chicago, IL, USA) was utilized to conduct the statistical analysis.

## 3. Results and Discussion 

### 3.1. Zeta-Potential, Solubility, and Colour Profiles of Pig Brain Proteins at Various pHs

In general, protein solubility at alkaline or acid pH is used to produce PPI, which is then precipitated at the pI [11]. Figure 1a,b demonstrate the zeta-potential profile and solubility characteristic of pig brain proteins at various pH levels. Proteins with positive charges were found at pH levels ranging from 1 to 4.5, with pH 2 having the highest charge frequency (*p* < 0.05; Figure 1a). At pH 5.5–14, proteins contained negative charges, with pH 10 having the highest charge frequency (*p* < 0.05; Figure 1a). The zeta-potential was near 0 at pH 5 (Figure 1a), while the pI of most pig brain proteins was around pH 5. The solubility of pig brain proteins revealed a distinctive U-shape profile at various pHs (Figure 1b). The maximum protein solubility was observed at pH 1–3 in an acidic environment (*p* < 0.05). In an alkaline environment, protein solubility is highest at pH 12, followed by pHs 11.5 and 11 (*p* < 0.05). The minimum solubility was found at pHs 4, 4.5, and 5 (*p* < 0.05). The fact that pig brain contains a range of proteins caused the pH solubility profile to vary.

Table 1 shows the colour (*L**, *a**, and *b**), haem protein content, and appearance of protein solutions at various pHs. The lowest *L** value was discovered at pH 2 while the greatest *L** value was discovered at pH 13. Under alkaline conditions, the *L** values of protein solutions appeared to be higher than under acidic conditions. This could be due to the increased protein solubility in an alkaline environment (Figure 1b). At pHs 1–14, negative *a** values were obtained in all samples, while negative *b** values were detected at pHs 1–7 and 14. Positive *b** values were apparent at pHs 8–13.

The total haem protein content of the protein solutions was lowest at pHs 1–5 (~0.01 g/100 g), increased at pHs 5.5–13 (~0.2–0.3 g/100 g), and declined at pH 14 (~0.1 g/100 g). The colour of protein solutions was related to the haem protein concentration. At pHs 1–4.5, the protein solutions were clear, but became a bit turbid at pH 5, and even more turbid at pHs 5.5–14. The milky red solutions were discovered at pHs 6–11.5, while the solution became turbid grey around pHs 12–14. At varying pHs, the extractability, oxidative stability, and degradation of haem protein, as well as protein structural changes and solubility, and emulsified lipid content, may all influence the final colour of the pig brain protein solutions.

In terms of protein solubility and zeta-potential, pH 2 (acid version) and pH 12 (alkaline version) were employed to extract protein, with pH 5.0 being utilised for precipitation.

### 3.2. Moisture Content, Dry Matter Yield, and Residual Lipid Content

The moisture content of the Acid-PPI and Alk-PPI was 83.84% and 91.12%, respectively (Table 2). The alkaline-assisted method yielded more dry matter (32.18%) than the acid-assisted process (6.0%) (*p* < 0.05) (Table 2). Protein accounts for the majority of dry matter in the PPI, with lipid and ash making up the rest [15]. The occurrence of charge frequency in the protein molecules produced by pH adjustment, as well as the presence of phospholipids that may bind water, resulted in an increase in moisture content throughout the pH-shift process. Furthermore, protein loss occurred predominantly as a result of separating into the top emulsion phase and precipitation during the first centrifugation stage. The decreased dry matter yield in the acid-aided method compared with the alkaline-aided method was likely owing to the alkaline-treated proteins’ superior lipid emulsion-forming capacity; therefore, more protein and lipid were recovered in the first centrifugation. This was supported by the Alk-PPI’s greater residual lipid, phospholipid, and total haem protein concentrations (Table 2). The pH change may have resulted in a drop in lipid concentration, especially in acidic conditions (Table 1). Ground pig brain had a lipid content of 9.08 g/100 g. In Acid-PPI, over 90% of the total lipid content was removed, whereas in Alk-PPI, roughly 50% was eliminated. When the pH is altered away from the protein’s pI, neutral and polar lipids may be released from the mince, resulting in the elimination of inter- and intracellular lipids, according to Phetsang et al. [15]. The larger residual lipids—notably, phospholipid—in Alk-PPI may be emulsified with protein and water and so bind the water in the PPI more intensively than in Acid-PPI. This was the reason for the increased moisture content of the Alk-PPI.

### 3.3. Total Haem Protein Content and Colour 

The Acid-PPI and Alk-PPI had residual haem protein levels of 4.32 and 10.89 mg/g, respectively (Table 2). The acid- and alkaline-aided processes eliminated 67.12% and 17.12% of the haem protein, respectively, when compared with the initial haem protein level (13.14 mg/g). The leaching impact generated by the added water, as well as the capability to partition haem from globin, resulted in a decrease in total haem protein from pig brain during the pH-shift procedure [25,26]. During the pH-shift procedure, haemoproteins were eliminated into the first sediment and/or second supernatant, according to Abdollahi et al. [12]. The degradation of haem protein, which results in the removal of the soret peak, has previously been demonstrated under acidic pH [26]. As a result, the amount of total haem protein identified was decreased. Only 17% of haem protein was removed at alkaline environment, which could be related to the structural stability of haem protein in an alkaline condition. As a result, it can be recovered in the final PPI alongside other proteins. Brown met-derivatives can be produced by oxidising haem proteins with acid and alkaline during the pH-shifting process [12,26]. Furthermore, chemical events such as lipid peroxidation and the Maillard browing reaction may have generated the colouring of the final PPI due to remaining haem protein and its degraded products (e.g., haemin and free iron) [15]. The greater *L** and lower *a** and *b** values, which led to the higher whiteness of Acid-PPI, may be associated with the reduced residual haem protein level and total lipid content (Table 2).

### 3.4. Surface HPB

Table 2 shows the surface HPB values of Acid-PPI and Alk-PPI. The increase in surface HPB is linked to the exposed hydrophobic regions of protein molecules because the hydrophobic sections are generally found in protein interiors. According to the findings, Acid-PPI exhibited a higher HPB than Alk-PPI (*p* < 0.05), indicating that the former had undergone considerable denaturation. Protein unfolding uncovers hydrophobic amino acids, affecting the protein’s surface HPB. However, a proper alteration in surface HPB was required for excellent protein functionality (e.g., gel-forming ability and surface activity) [22]. Depending on the balance between protein–water and protein–protein interactions, accessible hydrophobic residues might create connections and enhance the gel or aggregation network when heated [22].

### 3.5. Oscillatory Dynamic Rheology

Rheology can be used to establish the physicochemical factors that connect to gelation, which is the foundation for texture development. The storage modulus (G’) was used to determine gel formation (Figure 2a). The creation of an elastic structure increased the sample’s stiffness, as shown by an increase in G’ [27]. The viscosity modulus (G”) represents the viscous behaviour of food (Figure 2b). The tan δ illustrates the distribution of viscosity relative to elasticity (Figure 2c) [22,28]. Figure 2 shows the dynamic viscoelastic behaviour of Acid-PPI and Alk-PPI as they transition from sol to gel at different temperatures. When heated, the two samples responded differently. As shown in Figure 2a, the G’ in Alk-PPI tended to remain constant when the temperature was less than 50 °C and quickly increased when the temperature was raised to around 68 °C. Due to stronger attractive forces, the G’ peaked to about 65–68 °C, implying the formation of a robust gel network (i.e., hydrophobic interaction and disulphide bond). The covalent bonds between unfolded proteins were most likely responsible for the establishment of a thermo-irreversible gel network [29]. Hydrophobic domains may have connected via hydrophobic–hydrophobic contact while sulfhydryl groups were oxidised, resulting in a disulphide bridge [29]. G’ fell rapidly after that. The drop in G’ was caused by the α-helix coil transition, which enhanced protein fluidity while decreasing viscoelasticity [22]; this describes why G” reduced (Figure 2b). In addition, this could also be due to the fact that the hydrogen bond breaks off at elevated temperatures [30]. The viscosity modulus (G”) curve looked a lot like G’ (Figure 2a). The G” trended in the same direction as the G’. At temperatures below 50 °C, the G” remained constant. G” then steadily increased to 68 °C before falling due to cross-linking [22]. The tan δ stayed unchanged at temperatures < 50 °C, then raised until it reached 53 °C, and remained constant between 53 and 68 °C (Figure 2c). When the temperature was between 50 and 68 °C, the protein structure reorganised and the disulphide bonds and hydrophobic contacts between the molecules were reinforced, resulting in greater rigidity but impaired mobility. This could explain why the tan δ faded as the temperature increased above 68 °C [31]. 

As all of the parameters declined constantly with increasing temperature, the sol–gel transition after thermal treatment could not be generated in the case of Acid-PPI (Figure 2). This was most likely owing to excessive denaturation of proteins during acid treatment, which can result in protein agglutination. This was implied by the high initial values of G’ (Figure 2a), G” (Figure 2b), and tan δ (Figure 2c). Hydrogen bonds, which are easily broken by high temperatures, may help to stabilise the aggregates. As a result, as the temperature rose, the G’, G”, and tan δ dropped. The highly denatured proteins will not form a gel during heating, so only some degree of aggregation should be achieved. Alk-PPI had significantly higher G’, G”, and tan δ values at 90 °C than Acid-PPI. The results were consistent with the textural qualities (see Section 3.6).

### 3.6. Gelling Properties

#### 3.6.1. Breaking Force, Deformation, Expressible Drip, and Whiteness

Protein gelation is a useful feature that can be used in a variety of food products. According to the findings, Alk-PPI can be gelled, whereas Acid-PPI cannot (Table 2). When heated, Acid-PPI agglutinated, resulting in the creation of a protein aggregate (Table 1). Alk-PPI had a soft gel with a breaking force of 185.59 g, a deformation of 1.48 mm, and an expressible drip of 9.06% (Table 2). Due to their incapacity to form gel, the breaking force and deformation of Acid-PPI aggregate were not apparent. The Acid-PPI aggregate also had higher expressible drip (17.11%). Severe denaturation could be found in Acid-PPI leading to poorer gel-forming ability. 

Heat treatment reduced the whiteness of cooked Acid-PPI and Alk-PPI (Table 2) when compared with noncooked samples (Table 1). The oxidation of remaining haem proteins due to oxidative instability during the pH adjustment, as well as the reorganisation of proteins into three-dimensional networks, may all contribute to the cooked samples’ diminished whiteness. The oxidation of existing haemoproteins caused the reduced whiteness of PPI gels from Pacific whiting [32]. From the results, the Acid-PPI aggregate was whiter than the Alk-PPI gel in terms of colour characteristics after heating (*p* < 0.05) (Table 2). Increased exudate can reflect more light, making the colour whiter. In addition, the alkaline condition may have caused darkening via the Maillard reaction. The darker colour of Alk-PPI gel might thus be attributed to brown met-haemoproteins and Maillard reaction products [15].

#### 3.6.2. TPA and Microstructures

TPA is one of the most widely used approaches for determining food textural properties [33]. Hardness, springiness, cohesiveness, gumminess, and chewiness were all higher in the Alk-PPI gel than in the Acid-PPI aggregate (*p* < 0.05; Table 2). The former’s greater textural features were validated by the results. The protein aggregate with water release has a weaker structure than the three-dimensional network of gel with water entrapment. 

The microstructure of the Alk-PPI gel and the Acid-PPI aggregate differed (Figure 3). Acid-PPI generated a spongy network of aggregate with larger void spaces than Alk-PPI gel, resulting in lower textural parameters and more squeezable fluid (Table 2). On the other hand, Alk-PPI created a gel with a thick surface and continuously packed networks (Figure 3), in which water could be incorporated. Interestingly, the Alk-PPI formed a gel with connected clusters of proteins and beadlike clusters shielded/filled by a lipid layer, and fewer voids, when there were more residual phospholipids (Table 1). This was correlated with the higher breaking force and deformation with lower expressible drip of Alk-PPI gel (Table 2). It has been revealed that Alk-PPI gave the gel a greater breaking force and deformation than Acid-PPI in the case of fish [15]. Partial denaturation of proteins during the pH-shift process can aid in the generation of a gel structure [34]. Proteins may expose appropriate reactive groups under alkaline conditions, which may then interact during heat treatment [25]. During the heat gelation of the Alk-PPI, a network of protein molecules has been reported to develop, leading to increased gel strength [25]. Furthermore, Undeland et al. [35] discovered that Acid-PPI had faster protein hydrolysis than Alk-PPI, resulting in decreased gel strength.

Although it has been shown that retained lipids have a negative influence on surimi’s ability to gel [36], phospholipid has been found to improve surimi’s gel-forming capacity. According to Panpipat et al. [23], the best concentration of lecithin for maintaining the texture and hampering lipid oxidation of surimi gel from bigeye snapper was 1 g/100 g. Thus, the gelling ability of Alk-PPI was due to partial unfolding of the protein combined with the stabilising effect of residual phospholipid.

#### 3.6.3. Lipid Oxidation

TBARS were used to evaluate lipid oxidation (Table 2). The results revealed that Acid-PPI had higher lipid oxidation than Alk-PPI after heating (*p* < 0.05). Despite the fact that the pH-shift approach can lower both prooxidants and lipids, the isolates’ leftover lipids are clearly more vulnerable to lipid oxidation. The acid-assisted method yielded more TBARS than its alkaline counterpart (*p* < 0.05). This was most likely owing to the haem proteins’ pro-oxidative capacity, which was triggered at low pH levels [37]. Although the Alk-PPI had larger residual lipid and haem protein contents than its acid counterpart (Table 2), the former’s lipid oxidation was lower. This was attributable to a higher phospholipid concentration in the sample, which can act as a natural antioxidant [23].

### 3.7. Interfacial Properties

In comparison with commercial proteins such as WPI, SPI, and egg albumen, the interfacial properties of Acid-PPI and Alk-PPI are presented in Figure 4. Figure 4a,b demonstrate the EAI and ESI of Acid-PPI- and Alk-PPI-stabilised o/w emulsions. The emulsion with the highest EAI was Alk-PPI, followed by Acid-PPI or WPI, egg albumen, and SPI (*p* < 0.05). The greatest emulsifying activity was most likely owing to adequate protein unfolding, which revealed correct HPB and a considerable amount of residual phospholipid content (Table 2). Partially unfolded protein was most certainly responsible for the adsorption at the oil droplet interface [13]. The favoured configuration seemed to be the α-helix, which was the most compact amphiphilic structure at the oil–water interface, according to Li Zhai et al. [38]. Unfolded structures with the right HPB will be much more successful at stabilising the interface than folded structures [39]. At the water–oil interface, unfolded proteins reorient and form a viscoelastic film, resulting in the development and stabilisation of an emulsion. The EAI of both PPIs was higher than that of gelatin derived from splendid squid (*Loligo formosana*) skin (EAI ~ 35 m^2^/g) [40]. However, the ESI of Alk-PPI, on the other hand, was lower than WPI, equivalent to SPI, and higher than egg albumen and Acid-PPI (Figure 4b).

In terms of foam ability (Figure 4c) and foam stability (Figure 4d), Alk-PPI demonstrated excellent foam ability and foam stability (*p* < 0.05) that was comparable with egg albumen (*p* > 0.05). Although Acid-PPI may produce a stable foam, it was shown to be inferior to Alk-PPI and other commercial proteins (*p* < 0.05). All of the examined proteins had greater than 100% foaming ability and their foam stability was excellent.

Overall, Acid-PPI demonstrated poorer emulsifying and foaming activity than Alk-PPI. This was most likely due to the proteins’ increased degree of denaturation after acid processing, as indicated by the much higher surface HPB value (Table 2). The degree of denaturation of Acid-PPI may increase during emulsion preparation, completely impairing the interfacial characteristics. PPI—specifically, Alk-PPI—can be employed as emulsifier and foaming agent in food formulations, according to the findings.

## 4. Conclusions

For food sustainability, pig brain can be used as a protein source. The pH-shift techniques, particularly the alkaline process, can be employed to recover gel-forming and surface-active proteins from the pig brain. Solubilisation and precipitation can both be conducted at pH 12 and pH 5.0, respectively. This procedure could be one of the strategies used in the sustainable meat industry to achieve the zero-waste principle and increase the usage of by-products. Alk-PPI may be employed as a protein substitute in food products since it could form a soft gel and had good emulsifying and foaming properties. According to its techno-functionality, PPI from pig brain can be applied as a proteinaceous ingredient in a variety of foods such as salad dressings, meat products, and alternative desserts. 

## Figures and Tables

**Figure 1 foods-11-00695-f001:**
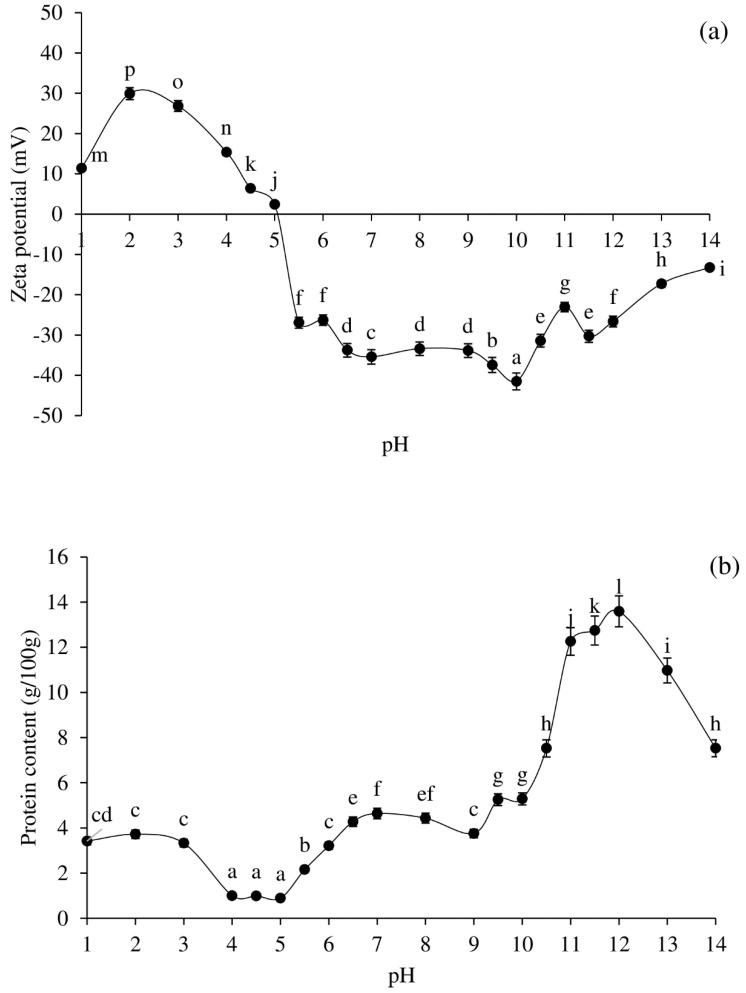
Effect of pH adjustment on zeta-potential (**a**) and protein solubility (**b**) profiles of pig brain protein. Bars represent the standard deviations from triplicate determinations. Different letters indicate the significant differences (*p* < 0.05).

**Figure 2 foods-11-00695-f002:**
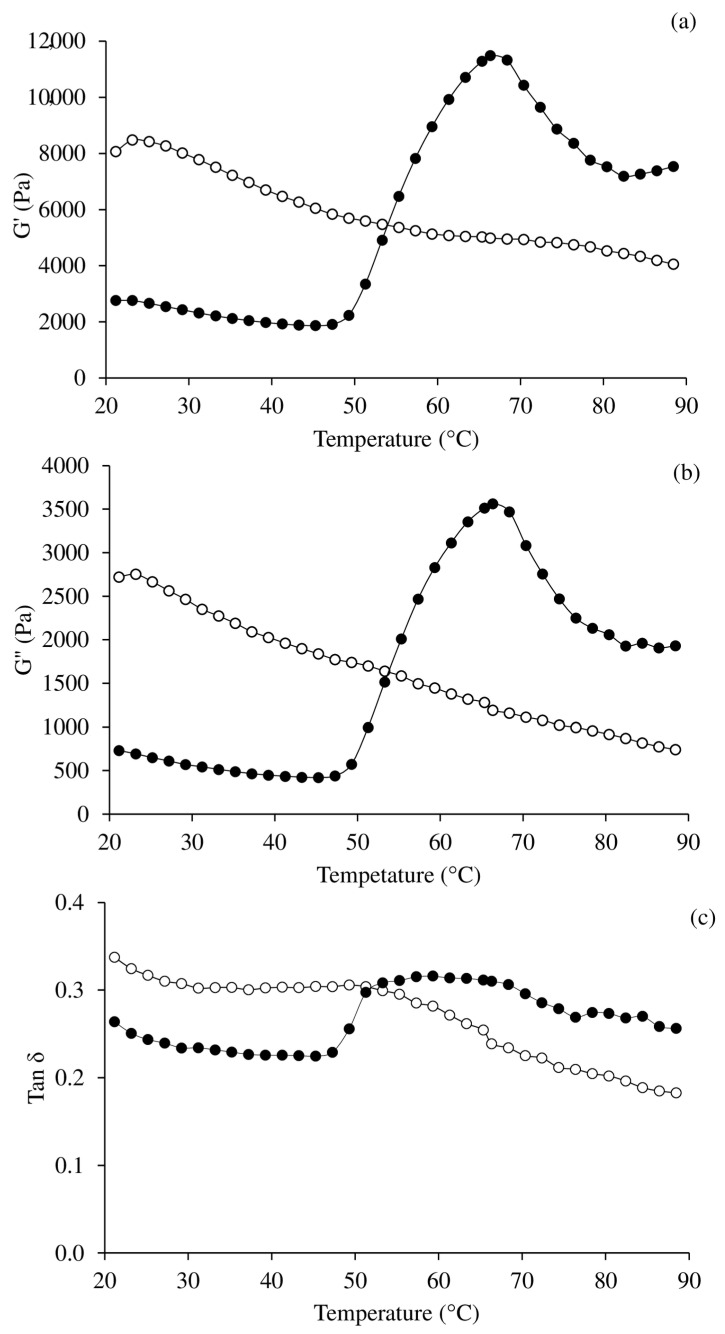
Changes in dynamic viscoelastic behaviour of acid-made protein isolate (○) and alkaline-made protein isolate pastes (●) from pig brain. The rheograms show elastic modulus, G’ (**a**); viscous modulus, G“ (**b**); and tan δ (**c**).

**Figure 3 foods-11-00695-f003:**
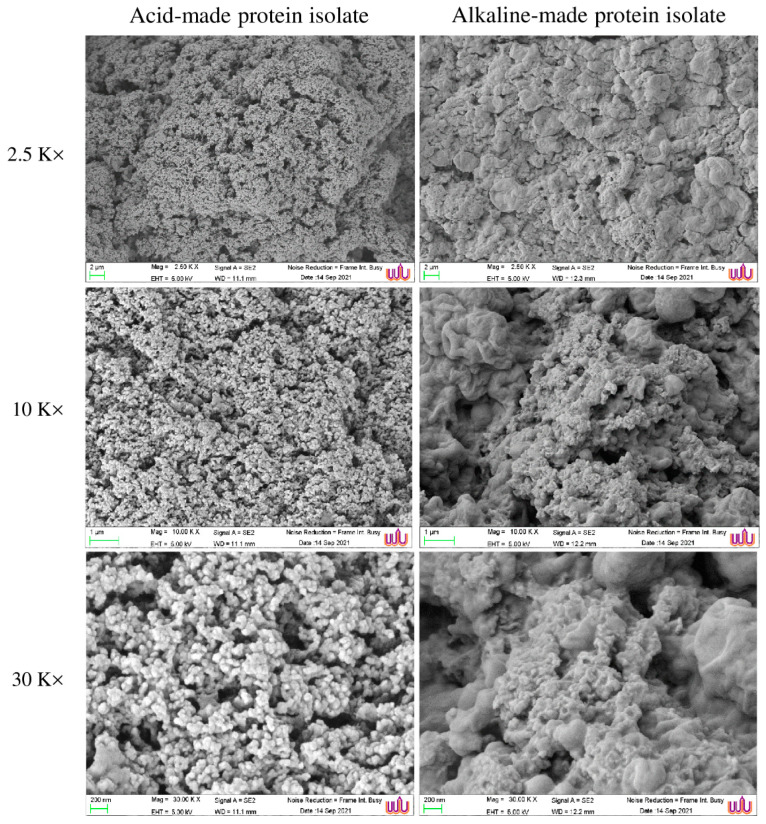
Microstructure of acid-made protein isolate aggregate and alkaline-made protein isolate from pig brain. Magnification: 2500×, 10,000×, and 30,000×. EHT: 5.0 kV.

**Figure 4 foods-11-00695-f004:**
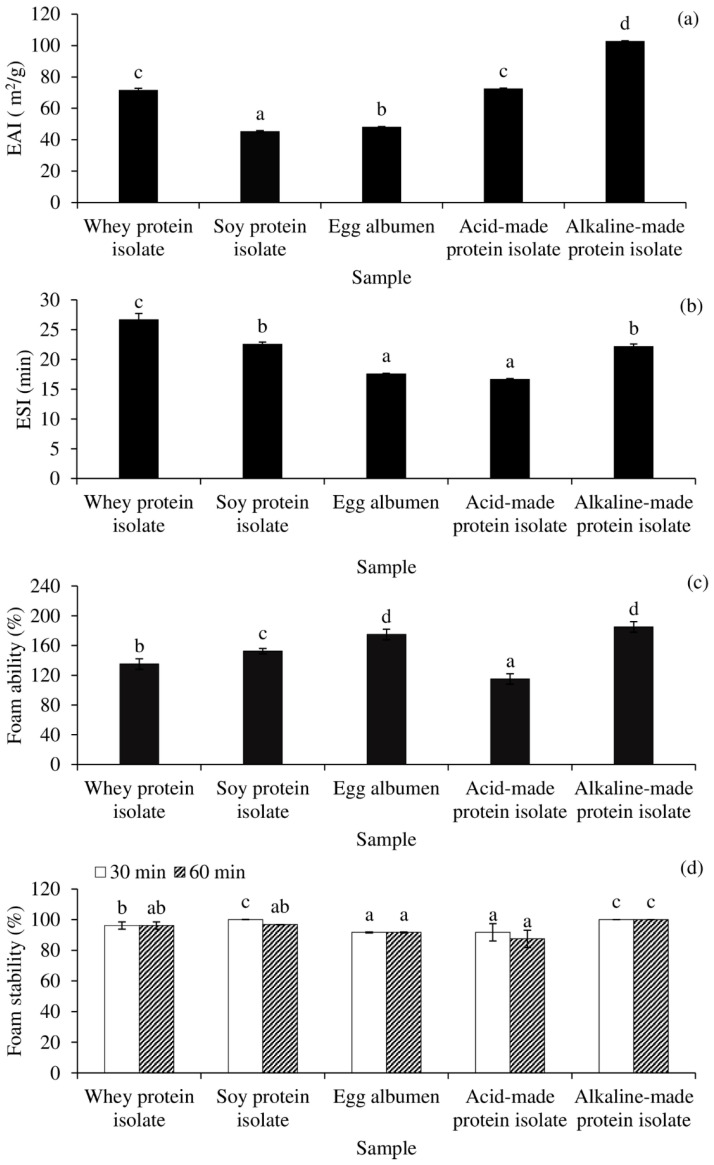
Emulsion activity index (EAI) (**a**), emulsion stability index (ESI) (**b**), foam activity (**c**), and foam stability (**d**) of acid- and alkaline-made protein isolates, in comparison with commercial proteins. Bars represent the standard deviations from triplicate determinations. Different letters indicate the significant differences (*p* < 0.05).

**Table 1 foods-11-00695-t001:** Effect of pH adjustment on colour (*L**, *a**, and *b**), total haem protein content, and appearance of pig brain protein.

pH	Colour	Haem Protein (g/100 g)	Appearance
*L**	*a**	*b**
1	23.19 ± 0.08 e	−0.34 ± 0.30 de	−2.04 ± 0.34 b	0.01 ± 0.00 a	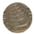
2	18.96 ± 0.53 a	−0.05 ± 0.23 e	−1.94 ± 0.24 b	0.01 ± 0.00 a	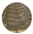
3	21.28 ± 0.65 b	−0.25 ± 0.23 de	−1.89 ± 0.10 b	0.01 ± 0.00 a	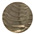
4	22.82 ± 0.17 de	−0.18 ± 0.15 de	−2.14 ± 0.12 b	0.01 ± 0.00 a	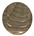
4.5	22.31 ± 0.55 cd	−0.30 ± 0.12 de	−2.31 ± 0.29 b	0.01 ± 0.00 a	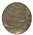
5	22.46 ± 0.49	−0.05 ± 0.13 e	−2.19 ± 0.15 b	0.01 ± 0.00 a	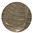
5.5	21.85 ± 0.32 c	−0.28 ± 0.11 de	−2.92 ± 0.04 a	0.15 ± 0.01 c	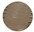
6	26.92 ± 0.27 ki	−1.35 ± 0.18 c	−1.50 ± 0.05 b	0.22 ± 0.01 d	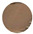
6.5	27.30 ± 0.07 m	−1.41 ± 0.55 bc	−0.66 ± 0.24 c	0.26 ± 0.01 g	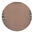
7	25.44 ± 0.46 g	−1.32 ± 0.10 cd	−0.73 ± 0.31 c	0.27 ± 0.01 g	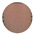
8	25.62 ± 0.41 gh	−1.62 ± 0.17 abc	0.33 ± 0.14 d	0.23 ± 0.00 de	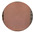
9	26.29 ± 0.18 ij	−1.60 ± 0.19 abc	0.61 ± 0.21 d	0.24 ± 0.01 ef	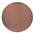
9.5	26.51 ± 0.18 jk	−1.61 ± 0.07 abc	0.99 ± 0.22 de	0.27 ± 0.01 g	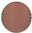
10	25.87 ± 0.26 ghi	−0.56 ± 1.66 de	0.51 ± 0.63 d	0.24 ± 0.00 f	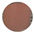
10.5	26.30 ± 0.44 ij	−1.55 ± 0.26 abc	1.85 ± 0.63 f	0.31 ± 0.00 i	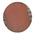
11	26.06 ± 0.15 hij	−2.22 ± 0.08 a	3.62 ± 0.43 h	0.29 ± 0.01 h	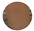
11.5	26.61 ± 0.18 jk	−2.16 ± 0.08 ab	2.96 ± 0.36 g	0.30 ± 0.00 hi	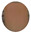
12	27.02 ± 0.17 lm	−2.29 ± 0.06 a	3.78 ± 0.23 h	0.30 ± 0.00 hi	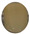
13	29.45 ± 0.16 n	−2.22 ± 0.04 a	1.28 ± 0.12 ef	0.25 ± 0.00 f	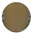
14	24.72 ± 0.12 f	−0.84 ± 0.13 cd	−2.89 ± 0.16 a	0.13 ± 0.00 b	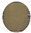

Values are given as mean ± SD from triplicate determinations. Different letters in the same column indicate significant differences (*p* < 0.05).

**Table 2 foods-11-00695-t002:** Quality characteristics of acid-made protein isolate and alkaline-made protein isolate from pig brain.

Parameters	Acid-Made Protein Isolate	Alkaline-Made Protein Isolate
Moisture content (g/100 g)	83.84 ± 0.10 a	91.12 ± 0.12 b
Dry matter yield (%)	6.00 ± 0.48 a	32.18 ± 1.14 b
Uncooked		
Appearance	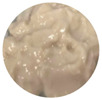	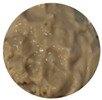
Residual lipid (g/100 g)	0.88 ± 0.23 a	4.56 ± 0.51 b
Residual phospholipid (mg/100 g)	1.32 ± 0.16 a	34.66 ± 0.04 b
Residual haem protein (mg/g)	4.32 ± 0.35	10.89 ± 0.15
Colour		
*L**	76.70 ± 1.01 b	66.31 ± 0.15 a
*a**	−1.16 ± 0.03 a	−0.56 ± 0.16 b
*b**	6.82 ± 0.09 a	10.85 ± 0.06 b
Whiteness	75.69 ± 0.95 a	64.60 ± 0.13 b
Hydrophobicity (BPB bound; μg)	102.69 ± 1.03 b	38.62 ± 0.27 a
Cooked		
Appearance	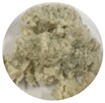	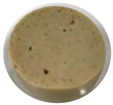
Breaking force (g)	ND	185.59 ± 1.01
Deformation (mm)	ND	1.48 ± 0.01
Expressible drip (%)	17.11 ± 1.69 a	9.06 ± 0.71 b
Whiteness	63.18 ± 1.36 b	55.99 ± 0.54 a
Texture profile analysis		
Hardness (N)	7.30 ± 0.10 a	8.03 ± 0.10 b
Springiness (cm)	3.38 ± 0.01 a	3.46 ± 0.01 b
Cohesiveness	0.11 ± 0.01 a	0.15 ± 0.01 b
Gumminess (N)	0.81 ± 0.02 a	1.21 ± 0.01 b
Chewiness (N.cm)	2.74 ± 0.20 a	4.22 ± 0.40 b
TBARS (mg MDA equivalent/kg)	0.74 ± 0.02 b	0.58 ± 0.01 a

Values are given as mean ± SD from triplicate determinations. Different letters in the same row indicate significant difference (*p* < 0.05). ND—not detectable because of no gel formed. BPB—bromophenol blue. TBARS—thiobarbituric acid reactive substances. MDA—malondialdehyde. The moisture, lipid, and total haem protein contents in ground pig brain were 79.96 g/100 g, 9.08 g/100 g, and 13.14 mg/g, respectively.

## Data Availability

Not applicable.

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
