# Peer review of "Recovery of Functional Proteins from Pig Brain Using pH-Shift Processes"

_foods, 2022, doi:10.3390/foods11050695_

Round 1
Reviewer 1 Report
The author used the pH-shift method to make protein isolate (PPI) from pig brain. The aqueous homogenate of ground pig brain was adjusted to either 12 (alkaline method) or 2 (acid method) using 2 M NaOH or 2 M HCl with steady stirring. The author compared quality features and functional aspects of acid-made protein isolate (Acid-PPI) and alkaline-made protein isolate (Alk-PPI). They further quantified the rheological, gelling, emulsifying, and foaming properties of the protein isolate.
The manuscript is interesting and well written and the authors have explained its work. The research work, if implemented, can help in waste utilization of the meat industry and value addition of pig brain. I have a few queries to clarify.
Comments
- Line 81: How has the solute: solvent ratio of 1:9 have been selected? What about using any other ratio, will it cause some effect on the protein yield?
- Line 101: Washing should be done after isoelectric precipitation to make sure that there is no residue of chemicals is present in the isolate.
- Line 144: Briefly describe the method used to determine interfacial properties.
- Line 174: Authors has written that a* value is negative at pH1-7 and 14 whereas in the table a* value is negative for all samples from pH 1-14
- Line 195: Also mention the crude protein content in extracted protein as well as in the sample.
Author Response
Reviewer 1
The author used the pH-shift method to make protein isolate (PPI) from pig brain. The aqueous homogenate of ground pig brain was adjusted to either 12 (alkaline method) or 2 (acid method) using 2 M NaOH or 2 M HCl with steady stirring. The author compared quality features and functional aspects of acid-made protein isolate (Acid-PPI) and alkaline-made protein isolate (Alk-PPI). They further quantified the rheological, gelling, emulsifying, and foaming properties of the protein isolate.
The manuscript is interesting and well written and the authors have explained its work. The research work, if implemented, can help in waste utilization of the meat industry and value addition of pig brain. I have a few queries to clarify.
Comments
Line 81: How has the solute: solvent ratio of 1:9 have been selected? What about using any other ratio, will it cause some effect on the protein yield?
Ans: According to prior studies, the sample to water ratio can be anything between 1:6 and 1:9. However, in this investigation, a ratio of 1:9 was used because we have shown that the higher the ratio, the better the ability to solubilize the proteins in the sample. References were also provided.
Line 101: Washing should be done after isoelectric precipitation to make sure that there is no residue of chemicals is present in the isolate.
Ans: It is a very nice suggestion. We will try this as a refinement process it in the future.
Line 144: Briefly describe the method used to determine interfacial properties.
Ans: Methods for determining interfacial properties were given.
Line 174: Authors has written that a* value is negative at pH1-7 and 14 whereas in the table a* value is negative for all samples from pH 1-14
Ans: Thank you very much. It was changed to “At pHs 1-14, negative a* values were obtained in all samples, while negative b* values were detected at pHs 1-7 and 14.”
Line 195: Also mention the crude protein content in extracted protein as well as in the sample.
Ans: Unfortunately, the PPI's protein content was not quantified. The dry matter yield was the only thing mentioned. However, protein is the most abundant dry matter in both PPI, with residual lipid and ash accounting for the remainder. As a result, the referenced phrase was added to indicate that protein is the primary dry matter in the PPI. “Protein accounts for the majority of the dry matter in the PPI, with lipid and ash making up the rest [15].”

Reviewer 2 Report
This study is investigating if pH-shift processing could be used to produce pig brain protein isolate (PI) which is important in terms of producing and characterizing pH-shift-made protein isolate (PPI) from pig brain as potential food ingredients. However, some corrections need to be made.
- The word "using" should be capitalized in the title.
- The writing of the city and zip code on Line 6 should be corrected.
- The citation of this report should be checked.
- The content and sentence order of the introduction paragraph between Line 33-50 is very similar to the introduction in the authors' previous published article, mentioned below. The introduction paragraph aforementioned should be checked again.
Chanted, J.; Panpipat, W.; Panya, A.; Phonsatta, N.; Cheong, L.-Z.; Chaijan, M. Compositional Features and Nutritional Value of Pig Brain: Potential and Challenges as a Sustainable Source of Nutrients. Foods 2021, 10, 2943. https://doi.org/10.3390/foods10122943
- Chemicals should be detailed in the "Material and Method" section.
- Grammar structure in Line 96 and Line 326 should be checked.
- The Biuret method [17], Chaijan and Undeland's method [18], Buege and Aust method [24], which are mentioned in the Materials and Methods section, should be briefly explained.
- Explanations of the terms L*, a*, and b* in Line 89 should be written.
- pH 12 and pH 2 should be written on Line 97 and 98.
- The Whiteness formula on Line 116 should be cited.
- 200 µl can be written numerically on Line 118.
- The device which is used for reading the absorbance of the supernatant on Line 123 should be mentioned.
- On Line 174, it is stated that at pHs 1-7 and 14, negative a* values were obtained in all samples. However, in Table 1, all the values of a* are negative.
- The size and resolution of the figures in Table 1 should be increased.
- Table 1 does not contain any information about lipid concentration. Therefore, Line 206 should be checked. Also, in Acid-PPI and Alk-PPI, the removed and eliminated lipid content is expressed as a percentage. The initial lipid content should also be mentioned in the sentence.
- It should be noted that the residual haem protein levels in Line 214 are found in (Table 2).
- In the Oscillatory Dynamic Rheology section, data about the Alk-PPI does not match the explanation of the graphs in Figure 2. It is stated that the G' in Alk-PPI tended to remain constant when the temperature was less than 50 °C and increased when the temperature was raised to around 68 °C. However, in Figure 2. alkaline-made protein isolates (which is shown with the symbol "o") decreased in all the graphs.
- Figures should be written in bold. Also, the figure name should be written without abbreviation in the paragraph.
- The citation on Line 266 should be checked.
- There is no information about the creation of protein aggregate when heated the Acid-PPI in Table 1. Line 297 should be checked.
- In Line 317, it should be noted that a comparison of hardness, springiness, cohesiveness, gumminess, and chewiness of Alk-PPI gel and Acid-PPI aggregate is found in (Table 2).
- In Figure 4, the abbreviation of emulsion stability index should be written.
- The writing of references 5, 10, 37 is incorrect.
- Although each result is discussed powerful in its own section with references, the discussion on the purpose of using protein isolates from pig brain as food ingredients can be expanded.
Author Response
Reviewer 2
This study is investigating if pH-shift processing could be used to produce pig brain protein isolate (PI) which is important in terms of producing and characterizing pH-shift-made protein isolate (PPI) from pig brain as potential food ingredients. However, some corrections need to be made.
The word "using" should be capitalized in the title.
Ans: Done.
The writing of the city and zip code on Line 6 should be corrected.
Ans: Done.
The citation of this report should be checked.
Ans: The citation style was double-checked using a journal guideline.
The content and sentence order of the introduction paragraph between Line 33-50 is very similar to the introduction in the authors' previous published article, mentioned below. The introduction paragraph aforementioned should be checked again.
Chanted, J.; Panpipat, W.; Panya, A.; Phonsatta, N.; Cheong, L.-Z.; Chaijan, M. Compositional Features and Nutritional Value of Pig Brain: Potential and Challenges as a Sustainable Source of Nutrients. Foods 2021, 10, 2943. https://doi.org/10.3390/foods10122943
Ans: We used Turnitin to look into the similarities and found that they were not plagiarized.
Chemicals should be detailed in the "Material and Method" section.
Ans: In the "Material and Method" section, the chemicals were detailed.
Grammar structure in Line 96 and Line 326 should be checked.
Ans: QuillBot, a paraphrase tool, was used to double-check the grammar structure in Lines 96 and 326.
The Biuret method [17], Chaijan and Undeland's method [18], Buege and Aust method [24], which are mentioned in the Materials and Methods section, should be briefly explained.
Ans: Biuret a well-established method. As a result, no additional information was provided. However, the methods of Chaijan and Undeland [18] and Buege and Aust [24] were described in detail.
Explanations of the terms L*, a*, and b* in Line 89 should be written.
Ans: Done.
pH 12 and pH 2 should be written on Line 97 and 98.
Ans: Done.
The Whiteness formula on Line 116 should be cited.
Ans: Done.
200 µl can be written numerically on Line 118.
Ans: In most cases, the sentence will not begin with a number. As a result, the word "Two-hundred" is spelled out at the start of the sentence.
The device which is used for reading the absorbance of the supernatant on Line 123 should be mentioned.
Ans: Done.
On Line 174, it is stated that at pHs 1-7 and 14, negative a* values were obtained in all samples. However, in Table 1, all the values of a* are negative.
Ans: Thank you very much. It was changed to “At pHs 1-14, negative a* values were obtained in all samples, while negative b* values were detected at pHs 1-7 and 14.”
The size and resolution of the figures in Table 1 should be increased.
Ans: The size of the figures cannot be increased due to the limitations of the Table size. The resolution of all figures, however, was sufficient to differentiate the various colors among samples.
Table 1 does not contain any information about lipid concentration. Therefore, Line 206 should be checked. Also, in Acid-PPI and Alk-PPI, the removed and eliminated lipid content is expressed as a percentage. The initial lipid content should also be mentioned in the sentence.
Ans: Line 206 refers to Table 2, not Table 1. In the footnote of Table 2, the initial lipid content was mentioned. The initial lipid content, however, was also included in the sentence. "Ground pig brain had a lipid content of 9.08 g/100 g."
It should be noted that the residual haem protein levels in Line 214 are found in (Table 2).
Ans: Done.
In the Oscillatory Dynamic Rheology section, data about the Alk-PPI does not match the explanation of the graphs in Figure 2. It is stated that the G' in Alk-PPI tended to remain constant when the temperature was less than 50 °C and increased when the temperature was raised to around 68 °C. However, in Figure 2. alkaline-made protein isolates (which is shown with the symbol "o") decreased in all the graphs.
Ans: We apologies for the symbol in Figure 2 being incorrect. The symbol "o" stands for acid-made protein isolate. Alkaline-made protein isolate is denoted by the dark filled sign "●". Thank you so much for the thorough review, which will help to avoid any miscommunication.
Figures should be written in bold. Also, the figure name should be written without abbreviation in the paragraph.
Ans: Done.
The citation on Line 266 should be checked.
Ans: Done.
There is no information about the creation of protein aggregate when heated the Acid-PPI in Table 1. Line 297 should be checked.
Ans: Line 297 refers to Table 2 in terms of gelling qualities, not Table 1.
In Line 317, it should be noted that a comparison of hardness, springiness, cohesiveness, gumminess, and chewiness of Alk-PPI gel and Acid-PPI aggregate is found in (Table 2).
Ans: Done.
In Figure 4, the abbreviation of emulsion stability index should be written.
Ans: Done.
The writing of references 5, 10, 37 is incorrect.
Ans: The references were double-checked in accordance with the journal's guidelines.
Journal Articles:
Author 1, A.B.; Author 2, C.D. Title of the article. Abbreviated Journal Name Year, Volume, page range.
Book Chapters:
Author 1, A.; Author 2, B. Title of the chapter. In Book Title, 2nd ed.; Editor 1, A., Editor 2, B., Eds.; Publisher: Publisher Location, Country, Year; Volume 3, pp. 154–196.
Although each result is discussed powerful in its own section with references, the discussion on the purpose of using protein isolates from pig brain as food ingredients can be expanded.
Ans: Overall, the applicability of pig brain protein isolate was recommended in the conclusion. “Alk-PPI may be employed as a protein substitute in food products since it could form a soft gel and had good emulsifying and foaming properties. According to its techno-functionality, PPI from pig brain can be applied as a proteinaceous ingredient in a variety of foods, such as salad dressings, meat products, and alternative desserts.”
